# Experimental Study of Irradiation of Thin Oxide and Mo/Si Multilayers by High Brightness Broadband VUV/UV Radiation and Their Degradation

Victor D. Telekh [1], Aleksei V. Pavlov [1], Daniil V. Kirillov [1], Evgeny V. Vorob'ev [1], Alexander G. Turyanskiy [2], Viacheslav M. Senkov [2], Petr A. Tsygankov [3], Freddy F. Parada-Becerra [3], Vladimir R. Vesnin [1] and Andrei S. Skriabin [1,*]

[1] Department of Power Engineering, Bauman Moscow State Technical University, 2nd Baumanskaya Street, 5, 105005 Moscow, Russia; telekh@bmstu.ru (V.D.T.); alekseipavlov@bmstu.ru (A.V.P.); kirillovdv@bmstu.ru (D.V.K.); evv@bmstu.ru (E.V.V.); vesnin.volodya@gmail.com (V.R.V.)
[2] Lebedev Physical Institute of the Russian Academy of Sciences, Leninskiy Avenue 53, 119991 Moscow, Russia; algeo-tour@yandex.ru (A.G.T.); senkov42@yandex.ru (V.M.S.)
[3] Department of Nuclear Physics and Astrophysics, School of Physics, Industrial University of Santander, Carrera 27 #Calle 9, Bucaramanga 680002, Colombia; piotrtsy@mail.ru (P.A.T.); freddy.parada@correo.uis.edu.co (F.F.P.-B.)
\* Correspondence: terra107@yandex.ru

**Abstract:** This study discusses the main features of the irradiation of prospective multilayer coatings by VUV/UV radiation from compressed plasma flows. Such radiation is characterized by a broadband spectrum and high brightness fluxes. Oxide and Mo/Si bilayers were used as the basis of the reflective multilayers for the visible and UV ranges. A gas-dynamic response from the irradiated surfaces was studied with schlieren photography. The properties of original and irradiated multilayers were described with ultra violet visible infrared spectroscopy (UV-Vis-IR), X-ray diffraction (XRD), X-ray reflectometry, scanning electron microscopy (SEM) and other techniques. Data on the degradation of optical properties and surface morphology were obtained.

**Keywords:** compressed plasma flows; VUV/UV radiation fluxes; thin multilayers; coating degradation



## 1. Introduction

Nowadays, oxide thin multilayers (based on $ZrO_2/SiO_2$ and $HfO_2/SiO_2$ pairs) are used as protective coatings, which are resistant to laser irradiation [1–3]. Such bilayers are prospective due to a high melting point and a low intrinsic absorption [1]. The operating wavelengths of such coatings lie in the range of middle UV, visible and near IR radiation [4,5]. $HfO_2/SiO_2$ polarizers for 1054 nm with low stresses are utilized in high-laser-damage-threshold coatings for the OMEGA laser system [6]. Post-processing by heat and radiation treatment can improve the properties of oxide multilayers. Some aspects of the influence of post-processing annealing on phase composition and transmittance for $HfO_2/SiO_2$ coatings deposited by electron-beam evaporation with ion beam assisting were reported [7]. As found, the doping of 25% $SiO_2$ annealed at 600 °C mirrors demonstrated a low absorption and scattering. Reflection and transmission of a $ZrO_2/SiO_2$ coating prepared by a modified low temperature sol–gel method were studied in [8]. The best mirrors deposited on glass substrate had 3.5 $TiO_x/SiO_2$ bilayers, 515 nm operating wavelength and 84% of maximum optical refection. Stresses in $ZrO_2/SiO_2$ multilayers (with a period of 3–9) deposited by electron evaporation were under discussion also [9]. Residual stress was up to −300 MPa. In the work [10] UV-irradiation treatment increased the refractive index for $ZrO_2/SiO_2$ 25-layer film. There is an interest in damage study for $HfO_2/SiO_2$ and $ZrO_2/SiO_2$ multilayers under high radiation loads also [11,12]. Radiation durabil-

ity of the mentioned oxide multilayers is about $\approx$20–32 J/cm$^2$ [13] under visible and IR laser radiation.

There is a need to study the stability of specified and other coatings under the exposure of UV quanta also [14]. Experimental modeling of degradation of Mo/Si, W/Si and other pairs [15–17] is also necessary under extreme radiation loads for the design of novel UV sources and their optical elements. Different aspects of radiation damage of materials without global phase and chemical transitions were highlighted earlier [18,19]. Such radiation defects are explained by the creation of Frenkel pairs and their associations.

Nowadays, high-pressure UV emitters based on compressed plasma flows [20,21] is also under close consideration [22,23]. As an example of such devices, it is possible to consider a coaxial magnetic plasma compressor (MPC) of the erosion type [24]. The generation of VUV/UV radiation occurs as a result of joint deceleration of the high-velocity plasma flows in a background gas and action of ponderomotive force. This leads to the formation of a high-temperature region (a plasma focus) with a brightness temperature of $\approx$40–100 kK [25]. Such emitters are characterized by a high brightness and extended dimensions of the emission zone [22]. The energy conversion efficiency into radiation is about $\approx$0.3 [26]. Another important feature of the MPC is the generation of the plasma flows with a complex inner structure that includes strong shock waves, turbulent vortexes, zones of chemical and photochemical reactions, etc. [27]. In this case, it is possible to carry out a complex impact on materials, including mechanical loads and radiation exposure. Thus, the MPC is indispensable for testing material stability under extreme conditions. The selection of a background gas in the chamber allows for the filtration of the radiation spectrum by a "cutting-off" of hard quanta [28]. In the case of inert gases, maximal energies are equal to the first ionization potentials I, e.g., $I_{Ne} \approx 21.55$ eV for neon. A small addition of air or oxygen decreases the maximal energies up to $\approx$6 eV that corresponds to an absorption in the Schumann–Runge bands [29].

We note that such plasma accelerator generates high energy fluxes of $10^{20}$–$10^{22}$ photon/s. As mentioned [30,31], characteristic values of energy are approximately up to $10^{10}$–$10^{11}$ photon/s per single pulse for synchrotron radiation. Our fluxes are higher than other low-pressure plasma emitters allow (up to $10^{17}$ photon/s for a single pulse) [32]. Besides, synchrotron radiation is strongly directed and creates mainly local high-temperature fields and stress gradients when irradiating. Moreover, a maximal energy $E_{max}$ for UV synchrotron irradiation is determined by a filter "cutting-off" in a beam line. For example, $E_{max} \approx 11$ eV for a LiF filter. Thus, there is a possibility to study UV-irradiation of materials under much more extreme conditions than for some UV applications (for example, a luminescence VUV spectroscopy [33]).

In the present study, we mean degradation as the damage of surface morphology and the forming of its defects (craters, cracks, etc.) [34] and sedimentation of fine metal fragments due to electrode erosion, and/or phase and chemical transitions that caused deterioration in optical properties, coating integrity failure and its exfoliation. Thermal effects can have a crucial influence on these processes, in our opinion.

However, we note a lack of data on the damage and degradation of the multilayers under such radiation fluxes. The multiplicity of the physical processes includes non-stationary heat transfer in thin films [35], physicochemical transformations [36], coating evaporation [11], uncertainties of optical properties in the UV/VUV range, etc. These circumstances dictate the need for close experimental studies of the processes, both gas-dynamic responses from the irradiated materials and material science aspects of the coating degradation.

## 2. Materials and Methods

A photo of the sample placement (for Mo/Si samples) in the chamber is presented in Figure 1. Generation of compressed plasma flows and the radiation fluxes were carried out with a high-current plasma accelerator (MPC) with a coaxial electrode unit 1 [22,25,26]. The electrodes were made from AISI 321 steel. The inner diameter was 6 mm, and the external

diameter was 34 mm. An ablative interelectrode dielectric insert was made from PTFE. The MPC was installed in a stainless steel vacuum chamber with a volume of $1.5 \times 10^{-2}$ m$^3$. Windows on the chamber flanges allowed the watching for the plasma generation and the gas-dynamic response above an irradiated surface. Studied samples 2 were installed inside the chamber in holder 3. The sample disposition was determined by the dimensions $l_1 \approx 38$–39 mm and $l_2 \approx 29$–30 mm. A pulsed low-inductance capacitor of 18 μF with a stored energy of 3.6 kJ was used.

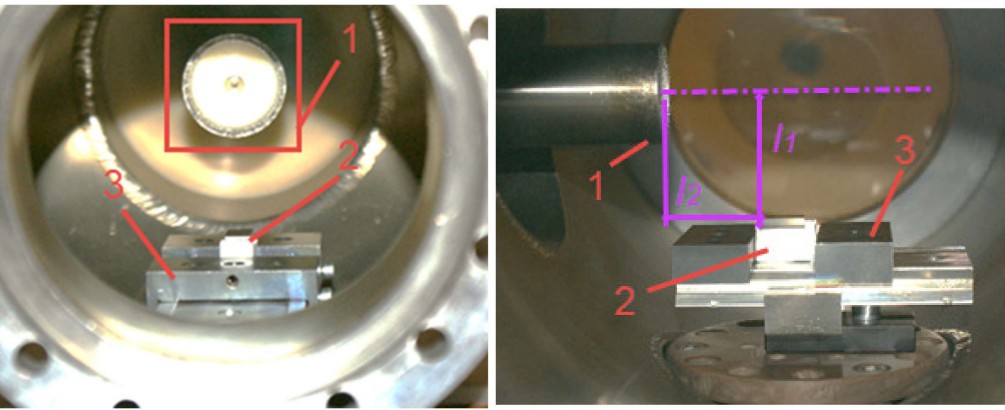

**Figure 1.** Scheme of sample (with Mo/Si coating) placement: 1–coaxial electrode unit, 2–sample and 3–sample holder.

The circuit was switched on by a thyratron (Pulsed Technology Ltd., Ryazan, Russia). Waveforms of the discharge current were recorded by a Pearson current monitor 110 and a Tektronix 2024b oscilloscope (Tektronix, Beaverton, OR, USA). Experiments were performed in pure neon, air and a neon/air mixture. A volumetric air concentration was ≈5% for the mixture. Pressure in the chamber was ≈300–400 torr for all studied cases.

Visualization and characterization of the gas-dynamic responses were carried out by laser schlieren photography in the bright field mode [37] for discharges in neon and the neon/air mixture. The wavelength of probing Nd:YAG laser beam was 532 nm. The images of inner structures of the generated flows were captured by a CCD Videoscan VS-285C camera (Videoscan Ltd., Moscow, Russia). The field of view for objects was 150 mm. The visualization technique allowed us to study gas-dynamic processes at temporal scales of ≈1–3 μs. The moment of laser triggering was detected by a photodiode. A detailed description of the optical diagnostics has been presented earlier [27,38].

The multilayers based on $ZrO_2/SiO_2$ and $HfO_2/SiO_2$ pairs were deposited by ion-assisted electron-beam evaporation on cylindrical fused quartz substrates (with a diameter of 5 cm). The specified coatings are considered as mirrors for the near IR radiation wavelength of 1054 nm. One side of the sample was turned to the MPC, and the opposite side was covered by a copper screen. Between the radiation exposures, the sample was turned 180° around. The Mo/Si coatings were deposited by magnetron sputtering on fused quartz bars ($15 \times 8 \times 3$ mm). For all these samples, radiation exposure in neon and air was studied. Specification of the prepared coatings is presented in Table 1.

The deposited coating (original and after single exposure) was characterized by a complex of diagnostic techniques. The phase composition was studied by XRD in the 2θ angle range of ≈10–145° (D8 and D2 Phaser, Bruker, Billerica, MS, USA) with an angle step of 0.02°. The XRD spectra were interpreted with the ICDD PDF-2 database. The inner structure of pairs was described by X-ray reflectometry (XRR, CDP systems, Moscow, Russia) with a Compleflex-5 X-ray reflectometer (CuK$_\alpha$ and CuK$_\beta$ radiations). The maximum 2θ angle was up to 5°. The reflectometry spectra were interpreted with the procedure [39]. Mechanical testing with a determination of hardness and Young's modulus was performed with the Oliver–Farr method (Nanovea nanoindentator, Nanovea, Irvine, CA, USA) at 4–5 points. The indentation load was 1 mN at the holding time of

30 s and the indentation depth of $\approx$0.1–0.9 µm. The loading and unloading rates were $\approx 3.3 \times 10^{-2}$ mN/sec. Profilometry was carried out with a KLA-Tencor D-600 device (KLA Corporation, Milpitas, CA, USA) (a diamond indenter with an apex angle of 45°). The load was 20 µN at the scanning speed of 0.2 mm/s. The measurement length was 0.08 mm. The data on roughness values were found. Electron microscopy was performed for the study of the surface morphology (TESCAN VEGA 3 XMU microscope; 30 kV; $10^{-3}$ Pa, Tescan, s.r.o., Brno, Czech Republic). Elemental composition was studied by EDX (AZTEC Energy Analysis System, Oxford Instruments, Abingdon, Great Britain). Measuring optical spectral characteristics was performed with Agilent Cary 7000 and Agilent Cary 300 spectrometers (Agilent, Santa Clara, CA, USA) in the wavelength range of $\approx$250–1300 nm with a step size of 1 nm.

**Table 1.** Specifications for prepared coatings.

| Parameter | Bilayer Type | | |
|---|---|---|---|
| | $HfO_2/SiO_2$ | $ZrO_2/SiO_2$ | Mo/Si |
| Bilayer number | 11 | 11 | 300 |
| Bilayer thickness (nm) | 316.8 | 306.0 | 11.4 |
| Thickness ratio | $\delta_{HfO_2}/\delta_{SiO_2} \approx 0.74$ | $\delta_{ZrO_2}/\delta_{SiO_2} \approx 0.68$ | $\delta_{Si}/\delta_{Mo} \approx 0.56$ |
| Total thickness (nm) | 3484.8 | 3366.0 | 3420.0 |

## 3. Results

Experimental studies allowed us to determine the main features of exposure of the VUV/UV fluxes on the oxide and Mo/Si multilayers. The data included information about both gas-dynamic processes above the irradiated surfaces and modification of the coatings (including their damage and degradation).

### 3.1. Features of Gas-Dynamic Response from Irradiated Oxide and Mo/Si Multilayers

The recorded waveforms of current I(t) and photodiode signal V(t) are shown in Figure 2. The maximal recorded current was $\approx$154 kA. We note that the sample irradiation occurred by powerful visible radiation during $\approx$3–4 µs at first. Further, the plasma heated, and the spectrum shifted to the VUV/UV range. The photodiode was not sensitive to this range. The laser triggering and image capturing occurred during an advanced stage of the discharge (the second half period I(t)) at $\approx$12 µs after the thyratron switching on. This was detected as a high peak of the photodiode signal.

It was found that only compressed and turbilized plasma flows from the MPC were visualized during discharges in air. No gas-dynamic structures (evaporation and shock fronts, etc.) were detected above both oxide and Mo/Si coatings. In the case of discharges in pure neon, the coatings based on $ZrO_2/SiO_2$ and $HfO_2/SiO_2$ pairs evaporated. The corresponding images are presented in Figure 3. The vapors were ionized by the hard quanta (with a maximal energy up to $\approx$21.55 eV).

As a result, the expanding plasma compressed the background gas and generated a shock wave. The shock wave front 1 and an interface (called a contact boundary) 2 between the plasma and the shocked gas were visualized. In the case of the exposure on the Mo/Si coating, strong evaporation and a contrast shock front was not detected, although an evaporation front 6 from the chamber wall was established. Weak fronts 1 and 2 from the sample were captured. It is important to note that the responses were registered without a mechanical impact of the shock front 3 from the MPC far from the irradiated sample 4.

### 3.2. Modification of $ZrO_2/SiO_2$ and $HfO_2/SiO_2$ Multilayers under Radiation Exposure

A photo of irradiated in neon and air $HfO_2/SiO_2$ multilayer is presented in Figure 4. The surface of such coatings was damaged under the exposure in neon. No visible damages were detected in the case of exposure in air. Parameters (roughness, mechanical properties and spectra) were measured at several random points in the area hatched by violet.

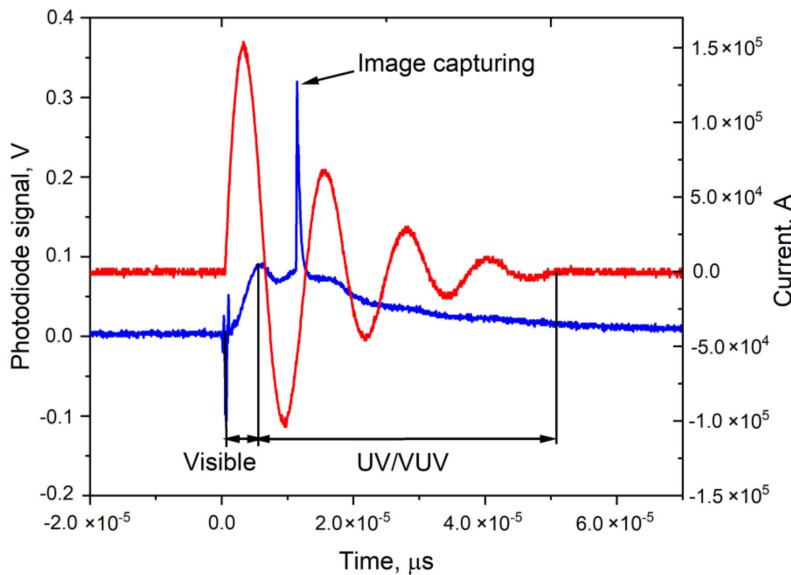

**Figure 2.** Waveforms of discharge current (red) and photodiode signal (blue).

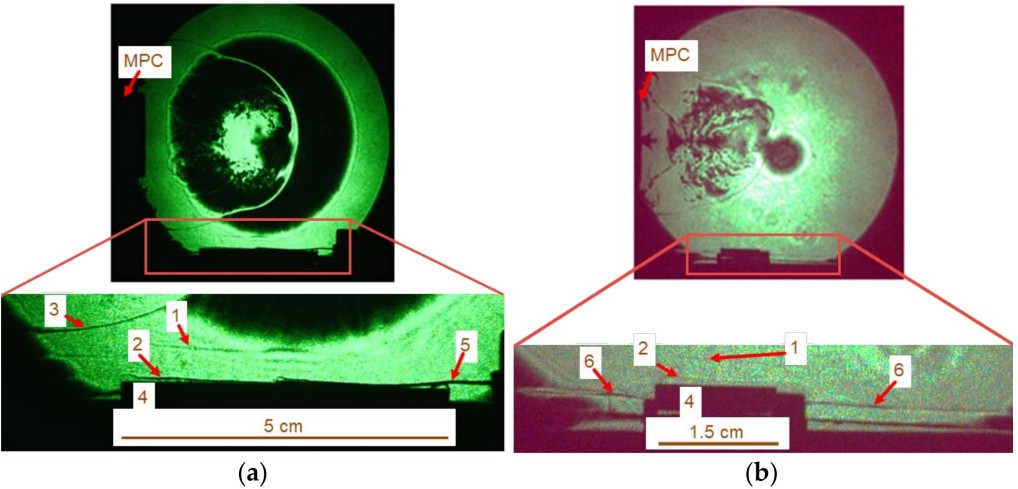

**Figure 3.** Schlieren images of captured flow structures during irradiation of $HfO_2/SiO_2$ (**a**) and Mo/Si (**b**) coatings in neon: 1—shock front from coating, 2—contact boundary, 3—shock front from MPC, 4—sample, 5—copper screen, 6—evaporation front from chamber walls.

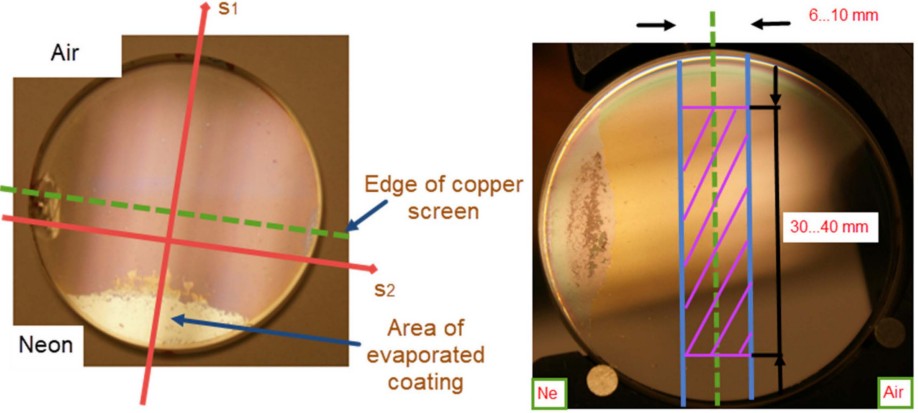

**Figure 4.** Photos of $HfO_2/SiO_2$ multilayer on fused quartz irradiated in neon and air. Parameters were measured in the hatched area. Roughness was measured along $s_1$ and $s_2$ directions.

The results of mechanical testing of the multilayers before and after exposures are presented in Table 2. The hardness of the uncovered substrate (fused silica) was $8.70 \pm 0.23$ GPa and its Young's module was $65.00 \pm 0.92$ GPa.

**Table 2.** Results of mechanical testing of $HfO_2/SiO_2$ and $ZrO_2/SiO_2$ multilayers before and after UV exposure.

| Parameter | $HfO_2/SiO_2$ Multilayer | $ZrO_2/SiO_2$ Multilayer |
|---|---|---|
| Hardness (GPa) | | |
| before | $5.34 \pm 0.29$ | $5.69 \pm 0.29$ |
| after | $4.90 \pm 0.22$ (air) | $4.10 \pm 0.10$ (air) |
| | $4.31 \pm 0.34$ (neon) | $3.14 \pm 0.43$ (neon) |
| Young's modulus (GPa) | | |
| before | $75.36 \pm 5.44$ | $81.38 \pm 6.91$ |
| after | $65.1 \pm 3.3$ (air) | $67.0 \pm 7.0$ (air) |
| | $76.0 \pm 11.0$ (neon) | $59.1 \pm 13.2$ (neon) |
| Roughness Ra (nm) | 0.49 | |
| before | $0.67$ (air, direction $s_1$) | |
| after | $0.58$ (air, direction $s_2$) | |
| | $0.43$ (neon, direction $s_1$) | |
| | $0.54$ (neon, direction $s_2$) | |

It was found that the roughness of the original coating was relativity slight (Ra = 0.49 nm for $HfO_2/SiO_2$ multilayer). The coating irradiated in air had larger Ra values of $\approx$ 0.58–0.67 nm. In the case of pure neon, Ra values were close to the initial values. The irradiated coatings had a larger scattering of Ra, and besides, it demonstrated a slight anisotropy of Ra, relatively a mutual disposition of the MPC and the sample (see Figure 4). For studied directions of scanning, the maximal deviation from the initial roughness was Ra = 0.67 nm in the $s_1$-direction under the UV exposure in air. In the $s_2$-direction perpendicular to the $s_1$ line, the deviation was less (Ra = 0.58 nm). For the case of irradiation in neon, the recorded Ra values were closer to the initial ones, but a small deviation was detected (Ra = 0.43 nm and Ra = 0.54 nm). The residual thickness in the area of intensive evaporation (near the substrate edge) was $\approx 10^3$ nm for the $HfO_2/SiO_2$ multilayer irradiated in neon.

The initial coating hardness was $5.34 \pm 0.29$ GPa (for the $HfO_2/SiO_2$ multilayer) and $5.69 \pm 0.29$ GPa (for the $ZrO_2/SiO_2$ multilayer). A decrease in the hardness was detected after the irradiation. Meanwhile, the maximal decreasing was found for irradiation in pure neon. Thus, the hardness values were $4.31 \pm 0.34$ GPa (for the $HfO_2/SiO_2$ multilayer; neon) and $3.14 \pm 0.43$ (for the $ZrO_2/SiO_2$ multilayer; neon). A similar trend, in general, was found for Young's modulus (see Table 2). The maximal reduction was found to be from $81.38 \pm 6.91$ GPa up to $59.1 \pm 13.2$ GPa for the $ZrO_2/SiO_2$ coating irradiated in neon.

Optical spectra were measured in both directions from the edge of the copper screen (see Figure 4). The reflectivity spectra of the original coatings, as well as the multilayers irradiated in neon and air, are presented in Figure 5. As it was found, exposure to UV/VUV radiation did not lead to any significant change of reflectivity at the studied points. All recorded spectra demonstrate the reflectivity maxima (up to R $\approx$ 98–100%) within the ranges of $\approx$340–390 nm and $\approx$920–1220 nm (for the $HfO_2/SiO_2$ coating); $\approx$350–400 nm and $\approx$940–1230 nm (for the $ZrO_2/SiO_2$ coating). The maximal deviation was detected for irradiation in neon, but its registered values did not exceed $\approx$2–3% at the studied locations. The reached values of R (up to $\approx$98–100%) at $\approx$1054 nm were close to the mirrors for which reflectivity was about $\approx$98% (for a $SiO_2/HfO_2/TiO_2$ coating [40]), $\approx$95% (for $HfO_2/SiO_2$ dichroic coating [41]) and $\approx$99% (for $ZrO_2/SiO_2$ thin coating [42]).

The XRD spectra of the coatings irradiated in neon and air are presented in Figure 6. As we found, the spectrum of the original multilayer (not presented in Figure 6) is close enough to the XRD patterns of irradiated coatings. The obtained spectra testified to the presence of monoclinic $HfO_2$ (PDF № 78-050) in the structure of the coatings. Its major

reflexes are indicated, too. No crystalline $SiO_2$ phases (quartz, cristobalite and tridymite) were reliably detected. An amorphous halo within the 2θ range of ≈16–37° was explained by an influence of the silica substrate.

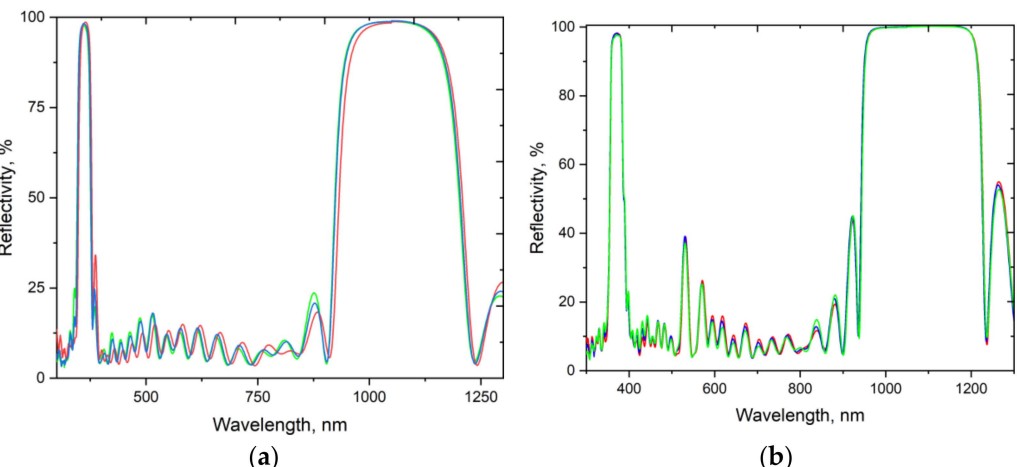

**Figure 5.** Reflectivity spectra of $HfO_2/SiO_2$ (**a**) and $ZrO_2/SiO_2$ (**b**) coatings: original (green), irradiated in neon (red) and irradiated in air (blue).

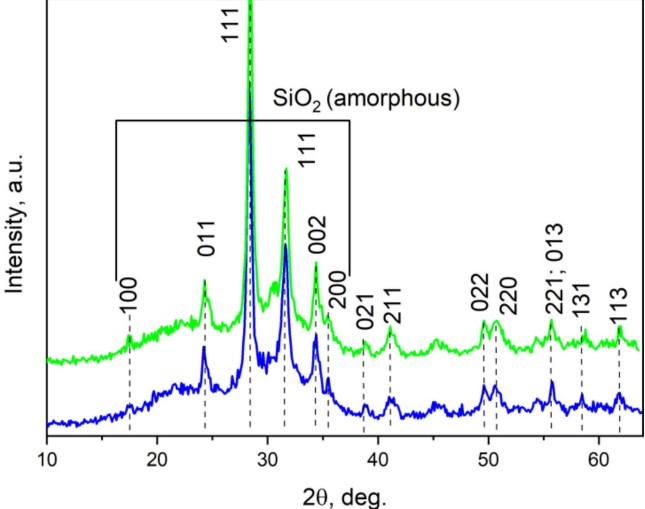

**Figure 6.** XRD patterns of $HfO_2/SiO_2$ coatings after exposure to UV/VUV radiation in neon (green) and air (blue).

### 3.3. Modification of Mo/Si Multilayers under Radiation Exposure

Optical photos of the irradiated in neon and air coatings are presented in Figure 7. Radiation exposure in air led to the formation of single defects of the coatings. Detailed studies by electron microscopy revealed the occurred changes in surface composition. The surface of irradiated in air coating had separate rounded defects (hollows with a size of ≈100 μm). These defects were formed by the interaction of molten debris drops from presumably the coaxial steel electrodes. Such defects were found with optical microscopy as bright glowing spherical-like objects as well. As it was shown by EDX, amounts of Fe (≈2.27–17.62%), Ni (≈0.29–2.37%) and Cr (≈0.25–3.16%) were found. The other found elements were Mo, Si and O. Significant changes, including film exfoliation, its multiple fracturing, formation of complex morphology, etc, were detected after the coating irradiation in pure neon. The multilayer surface also contained trace inclusions of Fe (≈0.21–1.48%) and Cr (≈0.16–0.47%) from the electrodes and the chamber wall. The elemental composition of the irradiated coatings (see Figure 7) is presented in Table 3.

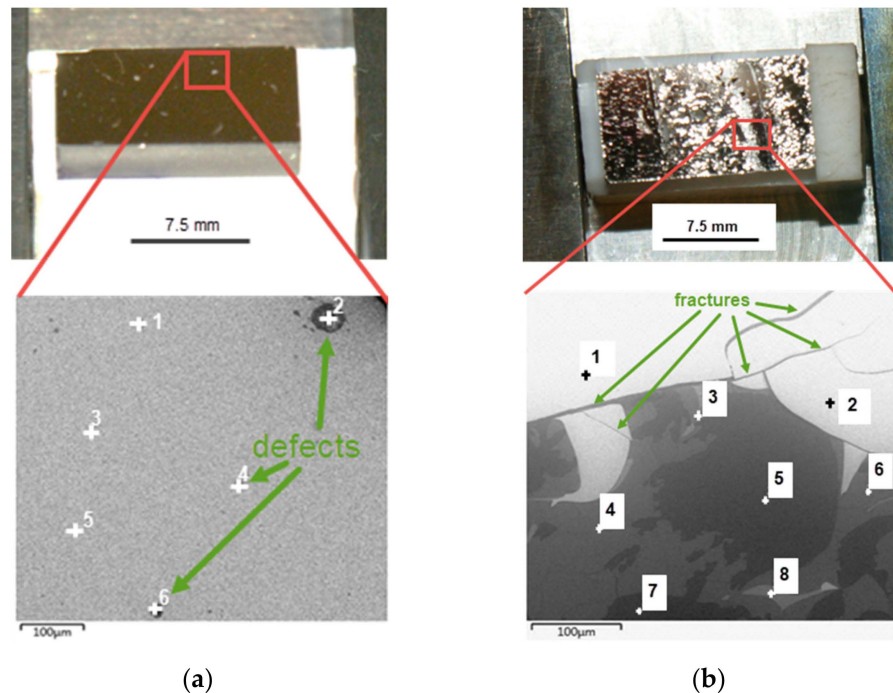

**Figure 7.** Optical and SEM photos of coatings irradiated in air (**a**) and neon (**b**).

**Table 3.** Elemental composition (at. %) on the surface of coatings irradiated in neon and air.

| Point | O | Si | Cr | Fe | Ni | Mo | Total |
|---|---|---|---|---|---|---|---|
| | | | Irradiated in Air | | | | |
| 1 | 5.38 | 24.06 | 0.25 | 2.39 | - | 67.92 | 100.00 |
| 2 | 15.67 | 18.82 | 3.16 | 17.62 | 2.37 | 42.36 | 100.00 |
| 3 | 3.72 | 24.32 | 0.56 | 2.11 | - | 69.29 | 100.00 |
| 4 | 14.13 | 19.60 | 1.81 | 9.21 | 1.76 | 53.49 | 100.00 |
| 5 | 3.63 | 24.25 | 0.86 | 2.38 | - | 68.88 | 100.00 |
| 6 | 9.78 | 23.08 | 0.54 | 2.27 | 0.29 | 64.04 | 100.00 |
| | | | Irradiated in Neon | | | | |
| 1 | 15.36 | 25.90 | 0.47 | 1.48 | - | 56.79 | 100.00 |
| 2 | 20.79 | 27.39 | 0.29 | 1.40 | - | 50.14 | 100.00 |
| 3 | 24.50 | 33.38 | 0.18 | 1.01 | - | 40.93 | 100.00 |
| 4 | 29.69 | 41.21 | 0.27 | 0.70 | - | 28.13 | 100.00 |
| 5 | 47.05 | 52.95 | - | - | - | - | 100.00 |
| 6 | 29.79 | 39.30 | 0.16 | 0.81 | - | 29.94 | 100.00 |
| 7 | 45.93 | 54.07 | - | - | - | - | 100.00 |
| 8 | 31.63 | 40.86 | 0.21 | 0.21 | - | 27.09 | 100.00 |

As shown in Figure 7a and Table 3, the highest ratio of Fe/Cr ($\approx$18/3–9/2) was detected for locations that corresponded to hardened debris steel drops (local defects or craters) from the electrodes. For irradiation, both air and neon, the Fe/Cr ratios are about the same on all other surfaces due to vapor condensation. We note that for irradiated in neon coatings increasing in oxygen content was caused by film exfoliating and denudation of the oxygen-rich silica substrate.

The XRD patterns of coatings are presented in Figure 8. As it was found, the original Mo/Si multilayers contained cubic Mo (PDF № 42-1120). No crystalline peaks of Si were

detected. A broad amorphous halo caused by the influence of the substrate was recorded for the 2θ range of <35°. Irradiation led to some phase and chemical transformations in the coatings. In the case of irradiation in air, cubic Mo was also detected. Besides, a one weak reflex of cubic Si (PDF № 27-1402) was detected. Significant changes were found for irradiation in neon. Essential decreasing of reflex intensities for Mo was found. Different phases of molybdenum silicides—cubic $Mo_3Si$ (PDF № 04-0814) and tetragonal $Mo_5Si_3$ (PDF №34-371)—appeared at the same time. Relatively broad reflexes of recorded phases testified to their small crystallite size.

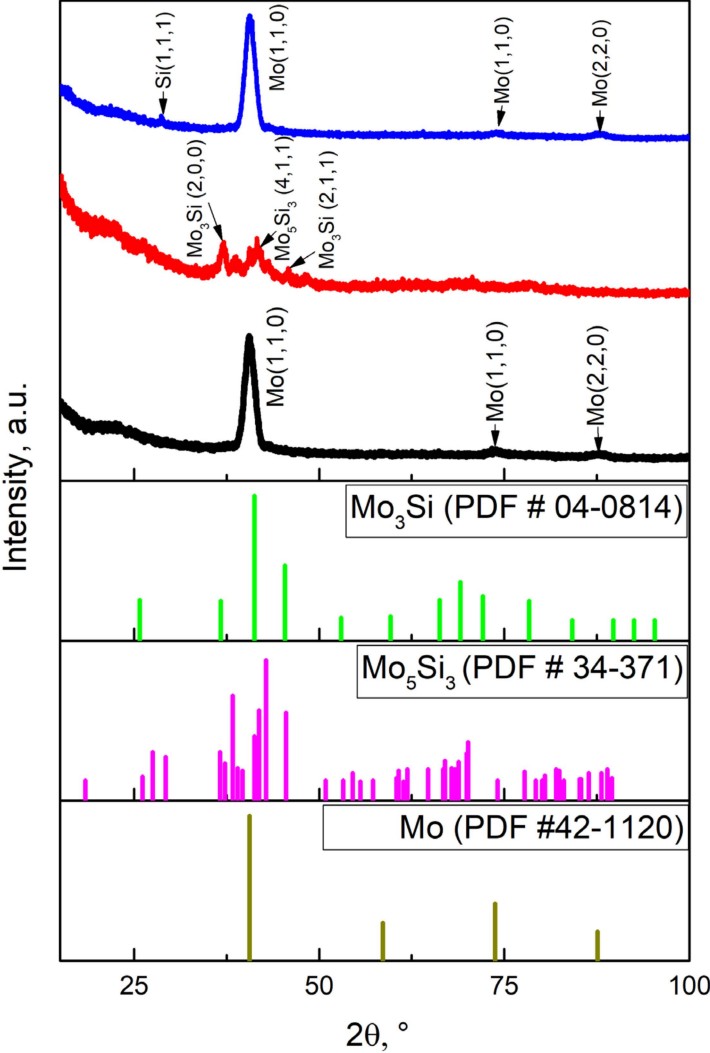

**Figure 8.** XRD patterns of coatings: original (black), irradiated in air (blue) and irradiated in neon (red). Reference patterns for Mo and molybdenum silicides are shown also.

Full widths at half maximum (FWHMs) of the Mo (1,1,0) reflex were $\Delta(2\theta)$ = 1.26–1.27° ($CuK_\beta$ radiation) for original and irradiated in air coatings. Measured using a Si single crystal, the FWHM of the device instrumental function was 0.05°. Estimated with the Debye–Scherrer formula, the Mo crystallite size was ≈7 nm. This value was near to the thickness of Mo layers in the Mo/Si bilayer determined by X-ray reflectometry. The complex morphology of the irradiated in neon coatings did not allow us to study their inner structure by XRR. On the other side, the reflectometry spectra testified to an almost complete identity of original and irradiated in air coatings. Original reflectivity was ≈0.56 ($CuK_\alpha$ radiation, $2\theta \approx 1°$). As it was found, a clarified thickness of the Mo/Si bilayer was 11.4 nm. Densities of Mo and Si layers were $\rho_{Mo}$ = 8.2 g/cm³ and $\rho_{Si}$ = 2.33 g/cm³. Their

roughness varied in the range of ≈0.9–1.3 nm. The external Si layer was covered by oxide silica film. Its thickness was 1.6 nm with a density of 2.2 g/cm$^3$. Irradiation in air did not lead to additional oxidation of the coating. Thus, the silica film served as a protector. Its roughness was about 0.6 nm. Reflectivity decreased up to ≈0.43 (CuK$_\alpha$ radiation, 2θ ≈ 1°) after the irradiation due to the formation of the hollows and craters in the coating. This is especially noticeable at small angles.

In general, it can be noted that UV/VUV exposure in air did not cause any significant changes in the thicknesses, densities, etc., for both oxide and Mo/Si coatings. The corresponding data on inner structures of some coatings obtained with XRR spectra are presented in Table 4. Because of significant damage from underexposure in neon, XRR was not suitable in this case.

**Table 4.** XRR data on inner structures of some coatings (original and irradiated in air).

| Coating | Parameter | | |
|---|---|---|---|
| | Thickness (nm) | Density (g/cm$^3$) | Roughness (nm) |
| Mo/Si bilayer | | | |
| SiO$_2$ (top) | 1.6 | 2.20 | 0.6 |
| Si | 4.1 | 2.33 | 0.9 |
| Mo | 7.3 | 8.2 | 1.3 |
| Fused quartz (substrate) | - | 2.50 | 0.5 |
| HfO$_2$/SiO$_2$ bilayer | | | |
| SiO$_2$ | 181.7 | 2.20 | 0.3 |
| HfO$_2$ | 135.1 | 9.68 | 1.4 |
| Fused quartz (substrate) | - | 2.50 | 2.2 |

## 4. Discussion

Absorbed energy (at the specific wavelength λ) in coating Q$_\lambda$ is calculated as [23]:

$$Q_\lambda = \tau \cdot A_\lambda \cdot I_\lambda \cdot \rho^{-1} z_\lambda^{-1} \cdot \mu \tag{1}$$

where τ ≈ 3–4 μs is a time of coating irradiation by the MPC discharge, A$_\lambda$ is an absorption coefficient, I$_\lambda$ is a radiation flux, ρ is a coating density, μ is its molar mass and z is an effective absorption depth. The value of z is determined by an attenuation coefficient α$_\lambda$ as z$_\lambda$ ≈ 4/α$_\lambda$. In the present study, the attenuation coefficient was estimated by the extrapolation of data calculated with the NIST Attenuation database [43]. A rough estimation of I$_\lambda$ is possible with data on the MPC radiance B$_\lambda$ [22,25,26] as:

$$I_\lambda \approx B_\lambda \cdot \Omega \cdot \lambda^{-1} \tag{2}$$

where Ω is a solid angle within which the coating is irradiated. In the present studies, Ω ≈ 10$^{-2}$–2 × 10$^{-2}$ str.

For an interpretation of the results, it is necessary to compare the values of Q$_\lambda$ and heats of phase and chemical transformations or their activation energies (i.e., the degradation thresholds). Considered in the present study, multilayers (oxide and Mo/Si) have fundamentally opposite thermophysical properties. HfO$_2$/SiO$_2$ and ZrO$_2$/SiO$_2$ pairs do not enter practically into chemical transformations under irradiation. However, phase transformations (e.g., evaporation) are possible. For oxide-based coatings with the upper silica layer (with a thickness of δ$_{SiO2}$ ≈ 180 nm), the first stage of degradation is associated with its evaporation as:

$$SiO_2(c) = SiO_2(g) \; [\Delta H_{ev1} \approx 590 \text{ kJ/mol}]. \tag{3}$$

For the other oxides evaporation heats are equal to

$$HfO_2(c) = HfO_2(g) \; [\Delta H_{ev2} \approx 610 \text{ kJ/mol}], \tag{4}$$

$$ZrO_2(c) = ZrO_2(g) \ [\Delta H_{ev3} \approx 810 \ kJ/mol]. \tag{5}$$

On the other side, systems based on Mo and Si layers can exothermically react under initial heating up to the melting temperature of silicon ($T_m \approx 1410 \ °C$) [44]. The corresponding activation energy $E_a$ of molybdenum silicide synthesis is varied from 139 to 167 kJ/mol [45]. Such exothermic processes are characterized by a complexity of transformation kinetics [46]. However, as previously reported [47,48], threshold energies can be less for reactive thin films compared with bulk reactants. Thus, all detected differences in the multilayer modification are explained by its chemical composition and the quanta energy. The enthalpies $\Delta H_{ev}$ were calculated with the NIST thermochemistry database [49]. The comparison of the threshold energies of degradation of oxide and Mo/Si multilayers and the absorbed energies for irradiation in air and neon is presented in Table 5.

**Table 5.** Values of degradation thresholds and absorbed in coatings energies for discharges in air and neon.

| Energy (kJ/mol) | HfO$_2$/SiO$_2$ or ZfO$_2$/SiO$_2$ Multilayers | Mo/Si Multilayer |
|---|---|---|
| Degradation threshold | 590 (silica evaporation) 610 (hafnia evaporation) 810 (zirconia evaporation) | 139–167 (MoSi$_2$ formation) |
| Discharge in air | 25–28 | 10–12 |
| Discharge in neon | 750–820 | 660–710 |

A temperature of the irradiated surface is calculated as [50]:

$$T_s = A \cdot I \cdot (\pi t k^{-1} \rho^{-1} C_p^{-1})^{0.5} + T_0. \tag{6}$$

where A is an integral absorption coefficient, I is an integral radiation flux, t is a discharge time and $T_0 \approx 300 \ K$ is an initial temperature. These values were calculated within the wavelength range of from $\lambda_{min}$ to $\infty$. The $\lambda_{min}$ value is determined by a spectrum "cutting-off". Thermal conductivity k of silica substrate was k = 1.4 W/(m·K). Density and heat capacity of the substrate were $\rho$ = 2.20 g/cm$^3$ and $C_p$ = 1.00 J/(g·K), respectively [51].

In the case of irradiation by soft quanta (the discharge in air, $\lambda > 200$ nm), the recorded reflectivity spectra (see Figure 5) testified to an effective reflectance (up to R $\approx$ 98–100%) in the near UV and near IR ranges. The original oxide multilayers are antireflective coatings in the visible range (with a transmissivity of T $\approx$ 80–90%). Thus, the main contribution to the heating was associated with irradiation by quanta with a relatively narrow wavelength range of $\approx$200–300 nm. The below estimations are presented for $\lambda$ = 200 nm. The measured value of absorption was $A_\lambda \approx 0.3$. The radiance was $B_\lambda \approx$ 50–60 MW·nm·cm$^{-2}$·str$^{-1}$. This corresponded to the radiation flux of $I_\lambda \approx 2.8 \times 10^{-3}$–$3 \times 10^{-3}$ MW/cm$^2$. For this case, the absorption length $z_\lambda$ was $\approx$33 nm. This value is less than $\delta_{SiO2}$. During the UV/VUV-irradiation (up to 3–4 µs), oxide coatings and the upper substrate layer were heated to a depth of $\approx$2–3 µm. Thus, in the case of coatings based on oxides, absorbance of energy occurred inside the upper SiO$_2$ layer. The absorbed energy was estimated with Equations (1) and (2) as $Q_\lambda \approx$ 25–28 kJ/mol. As we noted, modifications of oxide coatings were negligible under irradiation in oxygen-containing media due to low thermal and radiation loads. Slight declines of hardness and the Young module were caused by film annealing (see Table 2). On the other side, the energy was absorbed at the depth of about 3–4 bilayers for Mo/Si multilayers. The depth-averaged values of density and molar mass were $\rho$ = 5.94 g/cm$^3$ and $\mu$ = 96 g/mol. The reflectivity of Mo/Si multilayers demonstrates a complex behavior [52]. It is possible to highlight two characteristic regions of R: metallic reflection ($\lambda > 100$ nm) and multilayer resonance ($\lambda \approx$ 1–10 nm). Thus, absorption was $A_\lambda \approx 0.3$ for the considered wavelength. Thus, the absorbed energy was $Q_\lambda \approx$ 10–12 kJ/mol. The values of surface temperature were calculated with Equation (3).

In this case, A = 0.4 and I = 8–10 kW/cm$^2$. The temperatures were $T_s \approx 450$–550 K, that was too low to evaporate or totally melt coatings. Some decreases (from 54 to 43%) of reflectivity (CuK$_\alpha$ radiation) for Mo/Si coatings was explained by deposition of metal vapors (Fe, Cr, etc.) and their local damage by molten debris drops. An appearance of a weak reflex of Si (1,1,1) was caused by some crystallization due to the coating annealing (see Figure 8). As it was demonstrated by schlieren photography, heat and radiation fluxes did not evaporate the coatings.

When irradiated by hard quanta in neon ($\lambda > 60$ nm), the nearest edge to the MPC experienced high radiation loads. Due to the high absorption of oxide films (up to A = 1) the coating was partially evaporated at this location. The radiance was $B_\lambda \approx 170$–200 MW·nm·cm$^{-2}$·str$^{-1}$ for realized conditions. The radiation fluxes for the quanta with $\lambda = 60$ nm were about $I_\lambda \approx 5.7 \times 10^{-2}$–$7.0 \times 10^{-2}$ MW/cm$^2$. The calculated value of z for the upper silica layer was $z_\lambda \approx 80$ nm. Thus, the absorbed dose of energy at this location was $Q_\lambda \approx 750$–820 kJ/mol. With moving away from the MPC, the coating evaporation and degradation of its optical characteristics sharply decreased due to a reduction in radiation fluxes. No phase transformations were detected under irradiation of the studied oxide multilayers. In the case of irradiation of the Mo/Si multilayers, absorption was caused mainly in the upper 4–5 bilayers, wherein Mo layers absorbed almost all radiation. The values of $Q_\lambda$ were $\approx 660$–710 kJ/mol. With our estimations, the integral heat flux was I = 30–40 kW/cm$^2$. The corresponding surface temperature was $T_s \approx 2500$–2800 K. Such overheating exceeds the guaranteed temperatures of oxide evaporation and silicon melting.

Thus, exposure to UV/VUV radiation led to the initiation of synthesis reactions of molybdenum silicides in Mo/Si thin films. XRD patterns testified to the formation of two silicide phases (Mo$_3$Si and Mo$_5$Si$_3$). The detected broad silicide reflexes indicated small crystalline sizes due to a high disequilibrium of transformation kinetics. After the transformations, silicide films were cooling down, which led to their fracturing due to thermal stresses (see Figure 7b). The energies of hard quanta were enough for the formation of evaporation fronts and other gas-dynamic structures.

## 5. Conclusions

High-current coaxial accelerators generate compressed plasma flows that emit high-power broadband radiation, including the UV/VUV quanta. Selection of a background gas allows the spectrum filtration by a "cutting-off" of the hard quanta. Such plasma emitters can be used for the testing of the stability of thin multilayers under extreme conditions. In the present work, we studied the modification and degradation of oxide and Mo/Si thin multilayers by XRD, XRR, SEM, etc. Moreover, some features of the coating evaporation were visualized by laser schlieren photography. As found, irradiation slightly heated the coatings for discharge in air ($\lambda > 200$ nm). In this case, the main destruction mechanism is local damage by molten micron debris from the electrode and contamination by condensation of Cr and Fe vapors. Optical and morphological properties and phase composition changed slightly. Quite a different picture was observed for discharge in neon ($\lambda > 60$ nm). Exposure of quanta with the energy of $\approx 6$–22 eV caused the significant heating (up to $\approx 2500$–2800 K) of top layers that caused evaporation and chemical transformations. Evaporation was detected for oxide multilayers. Coatings based on Mo/Si bilayers reacted with the formation of molybdenum silicides and coating exfoliation.

**Author Contributions:** Conceptualization, A.S.S.; methodology, A.S.S. and V.D.T.; investigation, A.V.P., D.V.K., E.V.V., V.M.S., F.F.P.-B., V.R.V. and P.A.T.; data curation, A.S.S., A.G.T., P.A.T. and V.D.T.; writing—original draft preparation, A.S.S.; writing—review and editing, A.S.S., A.G.T. and P.A.T.; project administration, A.S.S. All authors have read and agreed to the published version of the manuscript.

**Funding:** The reported study was funded by RFBR and ROSATOM (No.20-21-00087).

**Institutional Review Board Statement:** Not applicable.

**Informed Consent Statement:** Not applicable.

**Data Availability Statement:** Not applicable.

**Acknowledgments:** The presented results have been obtained at large-scale research facilities "Beam-M" of Bauman Moscow State Technical University.

**Conflicts of Interest:** The authors declare no conflict of interest.

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
