# Peer review of "Experimental Study of Irradiation of Thin Oxide and Mo/Si Multilayers by High Brightness Broadband VUV/UV Radiation and Their Degradation"

_coatings, doi:10.3390/coatings12020290_

Round 1
Reviewer 1 Report
The submitted work reports the main features of irradiation of prospective multilayer coatings by the VUV/UV radiation. The mechanical testing results of HfO2/SiO2 and ZrO2/SiO2 multilayers before and after UV exposure are displayed in this study. Properties of original and irradiated multilayers were described with spectroscopy, XRD, XRR, and SEM. Finally, experimental data on degradation of optical properties and surface morphology were obtained and discussed. Overall, the scientific quality of the submitted manuscript is fine. The discussions based on the experimental results are sound. Several minor concerns are proposed herein for revision.
- The related scientific literature review on HfO2/SiO2 or ZfO2/SiO2 multilayers should be provided.
- The use of [h,k,l] is not often observed in the XRD patterns. Please use (h, k, l) to index the Bragg reflections in the XRD patterns.
- In Fig. 8, molybdenum silicides are observed in the XRD patterns. It is suggested to add XRD reference lines (from the referenced JCPD cards) in Fig. 8 for clarity.
- The authors claimed that X-Ray reflectometry (XRR) has been used in this work for structural characterization. However, the structural information obtained from XRR is limited in this work. For example, is the electron density of the coatings changed after irradiation? XRR can obtain the interface roughness, surface roughness, and density of the coatings before and after irradiation for a detailed structural analysis.
- The conclusion section should be enhanced to highlight the important findings of the work.
Author Response
Dear Colleague!
First of all, we would like to thank you for interest in the work.
Concerning you remarks we can note the following
1.“The related scientific literature review on HfO2/SiO2 or ZfO2/SiO2 multilayers should be provided.”
Ok. We’ll add the references in our introduction.
2.“The use of [h,k,l] is not often observed in the XRD patterns. Please use (h, k, l) to index the Bragg reflections in the XRD patterns”
Ok.
3.“In Fig. 8, molybdenum silicides are observed in the XRD patterns. It is suggested to add XRD reference lines (from the referenced JCPD cards) in Fig. 8 for clarity.”
Ok. We’ll present the reference patterns with the exception of Si because of its low reflex intensity in the coating. But the recorded peak Si(1,1,1) will be indicated.
4.“The authors claimed that X-Ray reflectometry (XRR) has been used in this work for structural characterization. However, the structural information obtained from XRR is limited in this work. For example, is the electron density of the coatings changed after irradiation? XRR can obtain the interface roughness, surface roughness, and density of the coatings before and after irradiation for a detailed structural analysis. “
Ok. We’ll add such data. But we note that obtaining of XRR data for irradiated in neon is difficult because of coating erosion.
XRR spectra are almost the same for original and irradiated in air films. See table 1 in the attachment file.
5. “The conclusion section should be enhanced to highlight the important findings of the work.”
Ok. A novel version is next.
High-current coaxial accelerators generate compressed plasma flows which emit a high-power broadband radiation including the UV/VUV quanta. Selection of a background gas allows the spectrum filtration by a “cutting-off” of the hard quanta. Such plasma emitters can be used for testing of stability of thin multilayers under extreme conditions. In the present work we studied modification and degradation of oxide and Mo/Si thin multilayers by XRD, XRR, SEM etc. Moreover, some features of the coating evaporation were visualized by laser schlieren photography. As found, irradiation slightly heated the coatings for discharge in air (λ>200 nm). In this case the main destruction mechanism is local damage by molten micron debris from the electrode and contamination by condensation of Cr and Fe vapors. Optical and morphological properties and phase composition changed slightly. Quite a different picture was observed for discharge in neon (λ>60 nm). Exposure of quanta with the energy of ≈ 6-22 eV caused a significant heating (up to ≈2500-2800 K) of top layers that caused they evaporation and chemical transformations. Evaporation was detected for oxide multilayers. Coatings based on Mo/Si bilayers reacted with formation of molybdenum silicides and coating exfoliation.
Best regards, authors

Reviewer 2 Report
Comments on v1
General comments
The paper describes the damage to reflective multilayer films, both oxide-based and metal-based, by high intensity UV/VUV radiation. However, there is insufficient data to substantiate all their conclusions. The authors claim that for oxide films, the structure and the reflectivity do not undergo significant changes. However, they also show that there is evaporation of the oxide film when exposed to Ne plasma. Evaporation of material will affect the thickness of material which will change the reflectivity (and probably also have a significant effect on the roughness). However, they suggest in figs. 4 and 5 that there is little difference to the roughness and essentially no difference to the reflectivity. It is not clear how this can be the case unless all the measurements have been taken from areas where evaporation did not occur. The authors have to indicate the position of the sampled areas for reflectivity and roughness. They mention XRR measurements but give no data. These measurements could be made on the irradiated samples and would also show up any thickness and roughness variations between the evaporated and intact regions. They have calculated an energy threshold above which evaporation of the top SiO2 layer will occur. However, significant evaporation will also require the evaporation of the HfO2 layers. How does the energy threshold for this compare? It would perhaps have been more useful to try to calculate an approximate temperature reached in the metal and oxide films to indicate what energy flux is required to create evaporation and/or melting. In section 4. Discussion, the authors state “In the case of irradiation of Mo/Si multilayers absorption caused mainly in upper 4–5 bilayers, wherein Mo layers absorbed all radiation while Si films were almost transparent.” How can Si layers be transparent to VUV radiation which is much higher than the band energy of Si?
Table 2 shows the composition of the Mo/Si films after irradiation. The samples irradiated in air show much higher Fe/Cr levels than those found in the samples irradiated in Ne. The authors should try to explain why this is so. If there is a difference in the number of molten droplets from the steel electrodes in the two cases why does that occur? Why are these droplets not seen on the oxide samples? Why is there a much lower oxygen content in the films irradiated in an air plasma where there would be a lot of active oxygen compared to the Ne plasma where there should be little oxygen?
The Conclusion is very sketchy, It does not give a good summary of the results.
In section 2. Materials and methods, the film construction in terms of bilayer thickness and number of bilayers is given. However, it is given in a different format for the oxide and metal layers. For clarity, for both types the bilayer thickness, number of bilayers, total thickness and ratio of layer thickness δA/δB should be given.
The paper says in section 2 “The maximum aperture of objects was 200 mm….” Should this read “The maximum aperture of the objective lens was 200 mm”? Please clarify.
In section 4, the paper says “On the other side, systems based on Mo and Si layers can react solid flame under an….”. I do not understand the phrase “react solid flame”. Please clarify.
In section 4, it states “When irradiated by hard quanta in neon (λ > 60 nm), the nearest edge to the MPC was undergone by high radiation loads”. This should read “When irradiated by hard quanta in neon (λ > 60 nm), the nearest edge to the MPC experienced high radiation loads”
On a point of English language, the authors on a number of occasions have used constructions such as “allowed to determine” or “allowed to study”. Although it is clear what they mean, standard English usage does not do this: “allow(ed) is followed be a direct object, for example “allowed us to determine” or “allowed us to study”. If the authors do not want to use “us” in the sentence then it should be changed to “allowed the determination of” or “allowed the study of”.
Author Response
Dear Colleague!
First of all, we would like to thank you for interest in the work.
Concerning you remarks we can note the following
1.«The authors claim that for oxide films, the structure and the reflectivity do not undergo significant changes. However, they also show that there is evaporation of the oxide film when exposed to Ne plasma. Evaporation of material will affect the thickness of material which will change the reflectivity (and probably also have a significant effect on the roughness). However, they suggest in figs. 4 and 5 that there is little difference to the roughness and essentially no difference to the reflectivity. It is not clear how this can be the case unless all the measurements have been taken from areas where evaporation did not occur. The authors have to indicate the position of the sampled areas for reflectivity and roughness».
This remark concerns some features of surface modification under irradiation from compressed plasma flows. Despite extensive researches a pattern of the radiation dynamics and its interaction with surfaces is not full. We found that heat processes (including evaporation) are intensive on the substrate edge that is close to the MPC barrel. It's not entirely obvious due to a multiplicity of radiation processes at the MPC and at the extended shock front. So, the measured data are highly dependent on the sample location: high degradation and evaporation were at the nearest edge to the MPC. Weak changes were close to the sample centre. We agree that it is necessary to indicate the positions for measuring. This figure 1 is presented in the attachment file. Parameters (roughness, mechanical properties, spectra) were measured at several random points.
- «They mention XRR measurements but give no data. These measurements could be made on the irradiated samples and would also show up any thickness and roughness variations between the evaporated and intact regions»
We added some information about XRR in the table for the original and irradiated in air sample. But we must note that spectra of original and irradiated in air samples were almost the same. For irradiated in neon it was not possible to record XRR spectra because of the surface erosion. See table1 in the attachment file.
- “They have calculated an energy threshold above which evaporation of the top SiO2 layer will occur. However, significant evaporation will also require the evaporation of the HfO2 layers. How does the energy threshold for this compare? It would perhaps have been more useful to try to calculate an approximate temperature reached in the metal and oxide films to indicate what energy flux is required to create evaporation and/or melting”
Here we mean the failure occurs when at the top layer (i.e. silica) is evaporated. Anyway, the evaporation heats are close for the considered oxides (610 kJ/mol for HfO2, 590 kJ/mol for silica). In the case of ZrO2 this value is some higher (≈810 kJ/mol) but it is close to the absorbed energy (up to 780 kJ/mol). So, the presented estimations do not prohibit evaporation of the considered oxide pairs by VUV irradiation. On the other hand, the calculated values of the surface temperatures were ≈450…550 K for discharge in air and ≈2500…2800 K for neon. Such overheating exceeds the guaranteed temperatures of oxide evaporation and silicon melting.
- «In section 4. Discussion, the authors state “In the case of irradiation of Mo/Si multilayers absorption caused mainly in upper 4–5 bilayers, wherein Mo layers absorbed all radiation while Si films were almost transparent.” How can Si layers be transparent to VUV radiation which is much higher than the band energy of Si?»
As known [https://doi.org/10.1002/qua.10411], the major absorption of Si lies in the range of 2.5…5 eV. This is caused by a resonant interband absorption. In the case of shorter wavelength quanta attenuation spikes occur due to absorption by inner electron shells. But for neon discharge (the maximal energy is ≈21.6 eV=0.0216keV) such spikes were not detected. Calculated with NIST database [https://physics.nist.gov/] values is presented in figure 2 (see the attachment).
- «The samples irradiated in air show much higher Fe/Cr levels than those found in the samples irradiated in Ne. The authors should try to explain why this is so.»
As shown in Figure 7a and Table 2, the highest ratio of Fe/Cr (≈17/3…9/2) was detected for location corresponded to hardened debris steel drops (local defects) from the electrodes. For irradiation both air and neon the Fe/Cr ratios are about the same on all other surface, because of vapor condensation in films. Thus, Fe and Cr contents are higher in micron steel drops than for the thin metal films.
«Why are these droplets not seen on the oxide samples?»
Obliviously, a type of sample does not affect debris formation. Dielectric oxide coatings were not reliably visualized by our SEM (without additional conductive coating deposition). But such defects were found with optical microscopy as bright glowing spherical-like objects. See the attachment file (figure 3).
“Why is there a much lower oxygen content in the films irradiated in an air plasma where there would be a lot of active oxygen compared to the Ne plasma where there should be little oxygen?”
We note that for irradiated in neon coatings increasing of oxygen content was caused by film exfoliating and denudation of the oxygen-rich silica substrate.
- “The Conclusion is very sketchy, It does not give a good summary of the results.”
We agree. A novel version is next.
High-current coaxial accelerators generate compressed plasma flows which emit a high-power broadband radiation including the UV/VUV quanta. Selection of a background gas allows the spectrum filtration by a “cutting-off” of the hard quanta. Such plasma emitters can be used for testing of stability of thin multilayers under extreme conditions. In the present work we studied modification and degradation of oxide and Mo/Si thin multilayers by XRD, XRR, SEM etc. Moreover, some features of the coating evaporation were visualized by laser schlieren photography. As found, irradiation slightly heated the coatings for discharge in air (λ>200 nm). In this case the main destruction mechanism is local damage by molten micron debris from the electrode and contamination by condensation of Cr and Fe vapors. Optical and morphological properties and phase composition changed slightly. Quite a different picture was observed for discharge in neon (λ>60 nm). Exposure of quanta with the energy of ≈ 6-22 eV caused a significant heating (up to ≈2500-2800 K) of top layers that caused they evaporation and chemical transformations. Evaporation was detected for oxide multilayers. Coatings based on Mo/Si bilayers reacted with formation of molybdenum silicides and coating exfoliation.
- «In section 2. Materials and methods, the film construction in terms of bilayer thickness and number of bilayers is given. However, it is given in a different format for the oxide and metal layers. For clarity, for both types the bilayer thickness, number of bilayers, total thickness and ratio of layer thickness δA/δB should be given.»
Ok. We’ll present such data in table 2. See the attachment file.
- “The paper says in section 2 “The maximum aperture of objects was 200 mm….” Should this read “The maximum aperture of the objective lens was 200 mm”? Please clarify.”
It is an inaccuracy.
Should read as “The field of view of objects was 150 mm”. The field of view is determined by a window size on a flange. Minimal size is 150 mm. The mentioned earlier size of 200 mm corresponds to the maximal one.
- « In section 4, the paper says “On the other side, systems based on Mo and Si layers can react solid flame under an….”. I do not understand the phrase “react solid flame”. Please clarify.»
Thin films are prospective systems for study of exothermic chemical processes of synthesis [see doi: 10.1016/j.tca.2007.12.011]. This phenomenon is called solid flame combustion or self propagation high temperature synthesis [10.1002/adem.201701065].
The phrase should be read as “On the other side, systems based on Mo and Si layers can exothermically react under…”
- “In section 4, it states “When irradiated by hard quanta in neon (λ > 60 nm), the nearest edge to the MPC was undergone by high radiation loads”. This should read “When irradiated by hard quanta in neon (λ > 60 nm), the nearest edge to the MPC experienced high radiation loads”
Yes.
- “On a point of English language, the authors on a number of occasions have used constructions such as “allowed to determine” or “allowed to study”. Although it is clear what they mean, standard English usage does not do this: “allow(ed) is followed be a direct object, for example “allowed us to determine” or “allowed us to study”. If the authors do not want to use “us” in the sentence then it should be changed to “allowed the determination of” or “allowed the study of”.
We’ll polish our text to eliminate the indicated occasions.
Best regards, authors

Reviewer 3 Report
This version does not look worthy and cannot be recommended for publication in this form and at least needs major revision.
- Line 6. Before the word spectroscopy, an explanatory word is necessary, describing what kind of spectroscopy was used.
- The abbreviation XRR needs some explanation.
- 1 paragraph: “ … experimental modeling of degradation” clarification is required about the degradation of which properties do you mean?
- Clarification is required, how the discussed problem can be compared with UV/VUV synchrotron irradiation? See recent papers in this case:
Pankratov et al, phys. stat. sol. (b) https://onlinelibrary.wiley.com/doi/10.1002/pssb.202100475
Lushchik et al, Nuclear Instrum Methods B https://doi.org/10.1016/j.nimb.2015.07.004
- Last paragraph of introduction. Here, solid discussions of the mechanisms of formation of point defects and VUV-UV photon stimulated desorption must be considered.
- 4. Is it possible to obtain comparative data on how the surface / near-surface structure changes during irradiation using luminescent methods and raman spectroscopy. Furthermore, corresponding 2D mapping in this case would be absolutely useful and necessary.
- 5 Give a comparison of the data presented here with the literature data on the reflection spectra of the corresponding dielectrics, HfO2, Al2O3 and SiO2.
Author Response
Dear Colleague!
First of all, we would like to thank you for interest in the work.
Concerning you remarks we can note the following
- “Line 6. Before the word spectroscopy, an explanatory word is necessary, describing what kind of spectroscopy was used.”
Ok. “…UV-Vis-NIR spectroscopy was used…”
- «The abbreviation XRR needs some explanation.»
Ok. “XRR (X-ray reflectometry)”
- « “ … experimental modeling of degradation” clarification is required about the degradation of which properties do you mean?»
We mean degradation of surface morphology (forming of hollows and fine metal fragments due to electrode erosion, roughness etc.) and/or its phase and chemical composition that caused deterioration in optical properties, coating integrity failure and its exfoliation. In fact, we studied some thermal aspects of the coating damage included contamination by debris, coating evaporation and chemical transformation under heating by irradiation.
- “Clarification is required, how the discussed problem can be compared with UV/VUV synchrotron irradiation?”
We note that our plasma accelerator generate enormous energy fluxes of 1020-1022 photon/s. As mentioned [https://onlinelibrary.wiley.com/iucr/doi/10.1107/S1600577513032931;https://onlinelibrary.wiley.com/iucr/doi/10.1107/S0909049509037236], characteristic values of energy is about up to 1010-1011 photon/s per a single pulse for synchrotron. Our fluxes are higher than other low-pressure plasma emitters allow (up to 1017 photon/s) [https://doi.org/10.1002/ppap.202100061] for a single pulse. Besides, synchrotron radiation is strongly directed and creates mainly local high temperature and stress gradients when irradiating. Moreover, maximal energy Emax for UV synchrotron irradiation is determined by “cutting-off“with filters. For example, Emax≈11 eV for LiF. In our case, “cutting-off“is carried out by a background gas. The sample is located directly in the chamber. For discharge in neon Emax≈21.6 eV. We note that our plasma compressor is a simpler system for such experiment tests. The present study was carried out under much more extreme conditions than luminescence VUV spectroscopy [https://onlinelibrary.wiley.com/doi/10.1002/pssb.202100475].
- “Last paragraph of introduction. Here, solid discussions of the mechanisms of formation of point defects and VUV-UV photon stimulated desorption must be considered.”
The found point defects are a result of electrode erosion, molten debris formation and their interaction with coatings. They are not “traditional” radiation defects [https://www.sciencedirect.com/science/article/pii/S0168583X15005820?via%3Dihub]. In fact, they are defects of surface morphology that can decrease coating quality. Now we think it's more appropriate to call them as “craters”, “hollows” etc. to avoid confusion.
- “ Is it possible to obtain comparative data on how the surface / near-surface structure changes during irradiation using luminescent methods and raman spectroscopy. Furthermore, corresponding 2D mapping in this case would be absolutely useful and necessary.”
As mentioned above, we studied morphology and phase transformation under UV irradiation. The used techniques were absolutely sufficient for our purposes and explanation of damage conditions. But in our future works we will certainly study processes of UV exposure by offered methods (Raman etc.).
- “Give a comparison of the data presented here with the literature data on the reflection spectra of the corresponding dielectrics, HfO2, Al2O3 and SiO2.”
Ok. We’ ll give such data. Besides, we’ll take into account all you your comments in the final version of manuscript.
Best regards, authors
Round 2
Reviewer 3 Report
The authors have successfully responded / reacted to all comments, so that the article can be recommended for publication